# A painful journey to antivenom: The therapeutic itinerary of snakebite patients in the Brazilian Amazon (The QUALISnake Study)

Joseir Saturnino Cristino[1,2], Guilherme Maciel Salazar[1,2], Vinícius Azevedo Machado[1], Eduardo Honorato[1], Altair Seabra Farias[1,2], João Ricardo Nickenig Vissoci[3], Alexandre Vilhena Silva Neto[1,2], Marcus Lacerda[1,2,4], Fan Hui Wen[5], Wuelton Marcelo Monteiro[1,2]*, Jacqueline Almeida Gonçalves Sachett[1,6]

1 Department of Medicine and Nursing, School of Health Sciences, Amazonas State University, Manaus, Brazil, 2 Department of Teaching and Research, Dr. Heitor Vieira Dourado Tropical Medicine Foundation, Manaus, Brazil, 3 Division of Emergency Medicine, Department of Surgery and Duke Global Health Institute, Duke University, Durham, North Carolina, United States of America, 4 Instituto Leônidas & Maria Deane, Fiocruz, Manaus, Brazil, 5 Bioindustrial Centre, Butantan Institute, São Paulo, Brazil, 6 Department of Teaching and Research, Alfredo da Matta Foundation, Manaus, Brazil

* wueltonmm@gmail.com

**Data Availability Statement:** All relevant data are within the manuscript and its Supporting Information files.

## Abstract

Access to antivenoms is not guarranteed for vulnerable populations that inhabit remote areas in the Amazon. The study of therapeutic itineraries (TI) for treatment of snakebites would support strategies to provide timely access to users. A TI is the set of processes by which individuals adhere to certain forms of treatment, and includes the path traveled in the search for healthcare, and practices to solve their health problems. This study aims to describe TIs of snakebite patients in the Brazilian Amazon. This study was carried out at the *Fundação de Medicina Tropical Doutor Heitor Vieira Dourado*, in Manaus, state of Amazonas, Brazil. The itinerary from the moment of the bite to the patient's admission to the reference unit was analyzed. Sample size was defined by saturation. After an exploratory survey to collect epidemiological variables, in-depth interviews were conducted following a semi-structured guide. Patients originated from rural areas of 11 different municipalities, including ones located >500 kilometers from Manaus. A great fragmentation was observed in the itineraries, marked by several changes of means of transport along the route. Four themes emerged from the analysis: exposure to snakebite during day-to-day activities, use of traditional therapeutic practices, and personal perception of the severity, as well as the route taken and its contingencies. Access to healthcare requires considerable effort on the part of snakebite patients. Major barriers were identified, such as the low number of hospitals that offer antivenom treatment, poor access to healthcare due to long distances and geographic barriers, low acceptability of healthcare offered in countryside, lack of use of personal protective equipment, common use of ineffective or deleterious self-care practices, late recognition of serious clinical signs and resistance to seeking medical assistance. Health education, promotion of immediate transport to health centers and decentralization of antivenom from reference hospitals to community healthcare centers in the Brazilian Amazon are more effective strategies that would to maximize access to antivenom treatment.

**Funding:** This research was funded by Fundação de Amparo à Pesquisa do Estado do Amazonas-FAPEAM (PAPAC 005/2019, PRÓ-ESTADO and Posgrad calls, to WM) and by the Ministry of Health, Brazil (proposal no. 733781/19-035, to ML). ML and WM are research fellows from CNPq. The funders had no role in study design, data collection and analysis, decision to publish, or preparation of the manuscript.

**Competing interests:** The authors have declared that no competing interests exist.

## Author summary

In the Amazon Region, when individuals are bitten by a venomous snake, they are obliged to seek care in hospitals located in the urban area in order to receive adequate treatment, i.e., the use of antivenom. Despite the fact that snakebites occur mostly in rural areas and riverine and indigenous communities, antivenom is not available in the health units located in these areas. Thus, this individual may take a long time to arrive at the hospital, because in addition to the great distance he/she must travel, there may be difficulties in obtaining transportation. In addition, individuals often use home treatments, believing that they will be effective in reversing the effects of the envenomation. This study was carried out through the application of questionnaires and interviews conducted with patients victims of snakebites treated at the *Fundação de Medicina Tropical Doutor Heitor Vieira Dourado*, which is a reference center for the treatment of tropical diseases, including snakebites, and is located in Manaus, capital of the state of Amazonas, Brazil. The patients came from 11 municipalities, and many came from very distant municipalities. The routes, in addition to being long ones, are very fragmented, and require different forms of transport in order to reach the reference hospital. In the places where they perform their daily activities, patients feel safe and do not use protective clothing. Many home treatment practices are used, including some that can be harmful, such as tourniquets. It is common for patients to seek help only when the clinical signs worsen, i.e., when severe pain, extensive edema and bleeding occurs. It is believed that if antivenoms were made available in health units closer to the patients' place of residence, there would be a decrease in cases of delay in treatment, and thus reduce the possibility of complications.

## Introduction

Snakebite envenomations represent a major burden on public health worldwide [1]. In 2019, there were 30,482 snakebite cases reported in Brazil, 13,601 of which were in the Brazilian Amazon region. These data show an uneven distribution of these events in the country, because, despite having a population of only 8.7% of the national total, 44.6% of the snakebites occurred in the Amazon region of Brazil. This equates to an incidence that is five times higher than the rest of the country [2]. Most cases occur in poor and rural settings where reports are incomplete and unreliable, and patients usually take a long time to reach healthcare services. The incidence rate of snakebites in the Brazilian Amazon is currently over 50 cases per 100,000 people/year [3], and the lethality rate is 0.6% [4]. Furthermore, snakebite envenomation also has a potential high economic impact for the individual and society [5]. The official number of cases and deaths is probably underestimated among the riverine and indigenous populations due to the difficulties in reaching health centers in order to receive medical assistance, since more than 30% of the patients were attended more than 6 hours after being bitten [4].

Despite technological improvements in the manufacture of antivenoms, several bottlenecks still exist in access to these immunobiologicals in poor regions in Africa [6–8], Asia [9] and Latin America [10, 11]. In Brazil, antivenom availability and accessibility is not uniform to the most vulnerable parts of the populations, i.e., populations that inhabit the remote areas of the Brazilian Amazon region. Rather than being distributed to rural and indigenous health facilities, antivenom treatment is only available in urban areas. Thus, better logistic infrastructure and professional training are issues to be addressed in order to reduce the high morbidity and mortality rates associated with snakebite envenomations [10, 11].

As a medical emergency, snakebite envenomation requires a quick response, with treatment using antivenoms, and preferably within first 6 hours of the bite [4, 12]. However, rural populations can take days to reach health care facilities [13]. Since the availability of life-saving antivenoms is still deficient in remote areas of the Brazilian Amazon region, a large proportion of patients choose to undergo traditional therapeutic resources. In some areas, traditional healers are still the primary source of healthcare for most of the rural populations in the Amazon. Plants, animals and mineral-based therapies, and spiritual healing are techniques used to treat, diagnose and prevent illnesses.

Some of these therapies include harmful procedures that can lead to snakebite complications, such as secondary bacterial infections in the affected limb [14], necrosis and compartment syndrome, which may lead to temporary or permanent functional losses in affected patients. Amputation, scarring, and muscular atrophy, which all affect mobility, are potential chronic outcomes. As a result, life-long disabilities greatly affect the victim's quality of life and rehabilitation is rarely available in such regions [15].

## Therapeutic itineraries

A therapeutic itinerary (TI) may be defined as is the set of processes by which individuals or groups choose, evaluate and adhere to certain forms of treatment [16]. According to Gerhardt [17], the concept of TIs includes the search for different forms of healthcare and describes and analyzes individual and socio-cultural health practices to solve the patient's illnesses. When individuals experience signs and symptoms considered to be abnormal, they find different ways to solve their health problems, and construct a set of plans, strategies and projects aimed at resolving the condition. These strategies are organized as a chain of successive or overlapping events, not necessarily predetermined, which form an articulated unit after completion, thus composing the TI. When investigated by the researcher, the TI will appear as an individual interpretation of the patient; a conscious attempt to return to the prior state of health and has the aim of giving meaning or coherence to the various fragmented acts along the itinerary. From the appearance of one or more physical or psychological symptoms and their recognition as such, the patient is confronted with a complex network of possible choices. The procedures for choosing, evaluating and adhering to certain forms of treatment are complex and difficult to be covered and understood if the context within which the individual is inserted is not taken into account. This is especially the case given the context within which the individuals are inserted and the diversity of possibilities available (or unavailable) in terms of health care for these populations with different characteristics [17].

According to Kleinman [18], most health care systems contain three subsystems within which the disease is experienced (Fig 1). The popular subsystem comprises the non-specialized field and is most often related to subjective interpretations. Self-care implies self-prescription and decisions on the use of a treatment in a relatively autonomous process, with an underlying influence of the family and social group on the subject's decision [19]. In the case of snakebites in the Amazon region, a series of traditional homemade procedures, such as the ingestion of preparations using vegetables or parts of animals, blessings and prayers, as well as self-medication with industrialized drugs are commonly used by patients before making the decision to seek out the health service [20, 21]. For the indigenous populations, the folk subsystem, which is based on spiritual rituals and practices carried out by shamans using medicinal plants, still constitutes the main component of healthcare. After a snakebite, the displacement of the indigenous patient to a hospital to receive antivenom, or in case of the need for surgical procedures, is a radical event. Some tribes may interpret venous punctures or surgical cuts as a possibility

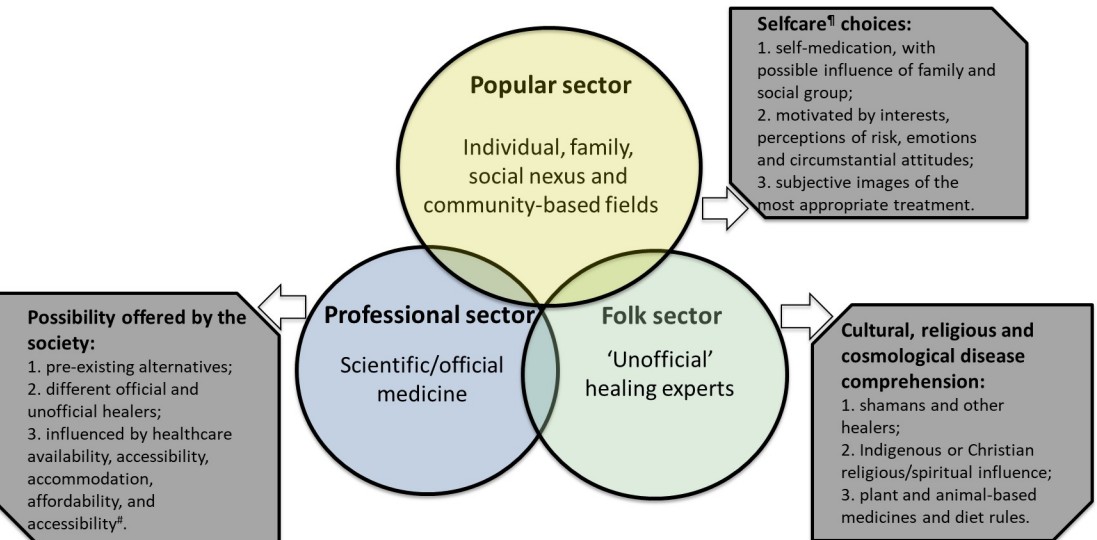

**Fig 1. Internal structure of the health system and factors affecting therapeutic itineraries.** Adapted from Kleinman [22].
#Definitions of the five dimmensions of healthcare are presented and discussed by Penchansky and Thomas [27]. ¶Self-care practices, representations and transactions with professional and folk sectors are detailed in Menéndez [23].

of inoculating a poison into the bloodstream, which generates bad blood, and is able to cause harm [22]. Additionally, another common belief is that the consumption of food produced outside the village and by other people could lead to their transformation into other beings [22, 23].

Regarding access to the healthcare service, specific dimensions include availability, accessibility, accommodation, affordability and acceptability [24]. The professional component of the system is usually located in the urban area of the municipalities, where antivenoms and ancillary treatment is available. In the case of snakebite envenomation in the Brazilian Amazon, even when these official therapeutic resources are accessible to the populations, traditional therapies are not fully replaced [14].

Inaccessibility to medical assistance forces patients bitten by snakes to travel long distances to the places where the antivenom is available. Other challenges include the lack of information on the social, cultural and economic aspects that prevent or impair the mobility of patients, as well as the availability of a health service. Knowledge of the TIs of snakebite patients would aid in the choice of appropriate strategies that guarantee care access at an opportune time and on a continuous basis, providing a link with the team of health professionals and, consequently, adherence to appropriate and timely treatment [25].

The objective of this study is to describe the itinerary from the moment of the snakebite to the patient's admission to the reference unit located in Manaus, in the Brazilian Amazon, where the antivenom was administered.

## Methods

### Ethics statement

The data collection for this study was carried out after approval by the Human Research Ethics Committee of the FMT-HVD (approval number 3.223.054/2019). All participants signed the consent form before participating. Patients were treated according the Brazilian Ministry of Health guidelines [26].

## Study subjects

This study was carried out at the *Fundação de Medicina Tropical Doutor Heitor Vieira Dourado* (FMT-HVD), located in Manaus, capital of the state of Amazonas, Brazil, and a reference hospital for the treatment of tropical diseases, including snakebites. The sample size was defined by the theoretical criterion of saturation [27], in which the data obtained from new participants began to show redundancy. A total of 30 patients aged ≥18 years treated at the FMT-HVD between October 2019 and March 2020 were thus included in the study.

## Study design

This is a cross-sectional study with a qualitative approach regarding the TI of patients presenting with snakebites at a reference center in the Brazilian Amazon. In this work, the itinerary from the moment of the bite to the patient's admission to the reference unit where the antivenom was administered is analyzed. Firstly, an exploratory questionnaire collected data of the sociodemographic characteristics of participants, and data concerning the snakebite envenomation, such as demographic information, signs and symptoms, anatomical site affected, use of traditional and complementary therapies, place of first hospital care, and time from snakebite to receiving medical care at FMT-HVD. Subsequently, in-depth interviews (IDIs) were conducted, which followed a semi-structured guide composed of 11 open questions (Table 1); the first three being considered rapport-building questions, that is, developed as a technique to create a connection of attunement and empathy with the interviewee in order to allow him or her to express themselves more freely and comfortably [28]. The additional questions allowed the interviewer to further investigate the participants' perception of the TI regarding the injury he/she had suffered. The questions were developed after discussions within the interdisciplinary research team, which has experience in assistance and research in snakebites, and were based on four thematic axes: 1. Therapeutic resources before antivenom treatment; 2. Individual trajectory; 3. Perception of severity; and 4) Contingencies during the trajectory. The interviews lasted approximately 40 min and took place in a quiet, comfortable room and were recorded for later transcription (S1 Text). The guide developed for the IDIs was previously tested and validated. The results and methods report follows consolidated criteria for reporting qualitative research [29]. All interviews were recorded using a Sony ICD PX240 4GB digital voice recorder. Interviews were transcribed to a file in Microsoft Word.

**Table 1. Thematic oral history interview guide.**

| Number | Question |
| --- | --- |
| Rapport-building questions | |
| 1 | Where are you from? |
| 2 | What do you do? (Work, study or stay at home) |
| 3 | What is your daily routine? |
| Therapeutic Itinerary-Related Questions | |
| 4 | Tell me what happened to you. |
| 5 | Tell me about the moment of the bite |
| 6 | What did you do right after the bite? |
| 7 | Did you take or do anything at home or before arriving at the hospital to treat the bite? What did you expect to happen? |
| 8 | Have you done this before? Who told you about it? |
| 9 | When a snakebite occurs, who do you seek in the community? |
| 10 | At what moment did you decide to go to the hospital? |
| 11 | How did you come from your home to here at the hospital? Were there any difficulties? |

## Data analysis

Data was presented according to time to medical assistance. Delay in access to treatment was defined as the interval between bite and arrival at the healthcare service being higher than 6 hours [3, 4]. Transcripts were assessed according to the technique of content analysis [30]. A pre-analysis phase was carried out through scanning and superficial reading of the interviews. Afterwards, a material exploration phase was performed using thematic framework analysis, and categories were thus created. These categories emerged during the analysis process and were discussed among the researchers for consensus. Clips of the conversations and the identified content were separated according to each interviewee. In the last step, logical consistency and the identified interpretation uncertainties from the conversations were solved by consensus among researchers for further interpretation.

## Results

### Characteristics of the participants

The characteristics of the study participants are presented in Table 2. One patient took 54 hours and another took 96 hours to reach the hospital.

Most of the patients were male (90%), of mixed ethnicity (87%), had ≤ 4 years of schooling (53%), and married or in stable relationships (53%). Participants were mostly involved in agriculture, hunting and fishing activities (67%). Snakebites were observed predominantly in lower limbs. Use of traditional therapies before hospital admission was reported by 67% of the patients. All participants presented pain, bleeding from bite site, edema and erythema. Secondary infections were reported in 40%, necrosis in 3% and osteomyelitis in 3% of the participants. Bleeding (3%) and renal failure (3%) were the only systemic manifestations reported. Snakebites were classified as mild in 37%, moderate in 57% and severe in 6% of the participants.

### Origin and physical itineraries of the study participants

The survey indicated that patients originated from the rural areas of 11 different municipalities (Fig 2). The municipalities with the highest number of cases were Careiro da Várzea (7 cases), Manaus (5 cases), Manacapuru (4 cases), Iranduba (3 cases), and Itacoatiara (3 cases). Of these municipalities, only Careiro da Várzea does not provide antivenom in its urban area [30]. Unexpectedly, patients from the municipalities of Nova Olinda do Norte, Itacoatiara, Novo Airão, Maués, and Novo Aripuanã, some of which are located between 200 and more than 1,000 kilometers from Manaus by land or river routes, were admitted at the reference center. Although late medical assistance was observed in patients from all the municipalities, we highlight that all cases from Autazes, Careiro Castanho, Maués, Nova Olinda do Norte, Novo Airão, Novo Aripuanã, and Itacoatiara were admitted to the hospital after a significant period of time had elapsed since the bite, which coincides with their distance from the capital.

Fig 3 reveals the itineraries adopted by each participant. Interestingly, most patients started their journey to the hospital immediately or shortly after the bite, which reveals an understanding of the need for antivenom treatment. Some individuals, however, only started the journey some time after the snakebite (one of them only after 27 hours), either after unsuccessful attempts to improve the signs and symptoms by using traditional treatments or due to lack of transportation at the time of the injury. A great fragmentation is observed in the itineraries, marked by several changes of means of transport along the route.

Fig 4 represents the itinerary of three study participants, with the route from where the envenomation occurred to the hospital, and shows the route taken along with its various fragments.

**Table 2. Characteristics of the study participants.**

| Variables | Total (n = 30) | |
|---|---|---|
| | N | % |
| **Sex** | | |
| Male | 27 | 90 |
| **Ethnic background** | | |
| Mixed/brown-skinned | 26 | 87 |
| Indigenous | 2 | 7 |
| White | 2 | 7 |
| **Age group (years)** | | |
| 18–29 | 10 | 33 |
| 30–39 | 8 | 27 |
| 40–59 | 10 | 33 |
| ≥60 | 2 | 7 |
| **Schooling (in years)** | | |
| ≤4 | 16 | 53 |
| >4 | 13 | 44 |
| **Marital status** | | |
| Unmarried/divorced | 14 | 47 |
| Married/stable relationship | 16 | 53 |
| **Occupation** | | |
| Agriculture, hunting and fishing | 20 | 67 |
| Others | 10 | 33 |
| **Anatomical site affected** | | |
| Foot | 14 | 47 |
| Leg | 6 | 20 |
| Ankle | 3 | 10 |
| Toe | 3 | 10 |
| Hand | 4 | 13 |
| **Used of traditional therapies** | | |
| Yes | 20 | 67 |
| **Type of snakebite¶** | | |
| *Bothrops* (pit vipers) | 30 | 100 |
| **Manifestations in the affected limb** | | |
| Pain | 30 | 100 |
| Bleeding from bite site | 30 | 100 |
| Edema | 30 | 100 |
| Erythema | 30 | 100 |
| Secondary infection | 12 | 40 |
| Necrosis | 1 | 3 |
| Osteomyelitis | 1 | 3 |
| **Systemic manifestations** | | |
| Bleeding | 1 | 3 |
| Renal failure | 1 | 3 |
| **Clinical evaluation** | | |
| Mild | 11 | 37 |
| Moderate | 17 | 57 |
| Severe | 2 | 6 |

¶*Bothrops* diagnosis was based on clinical-epidemiological criteria.

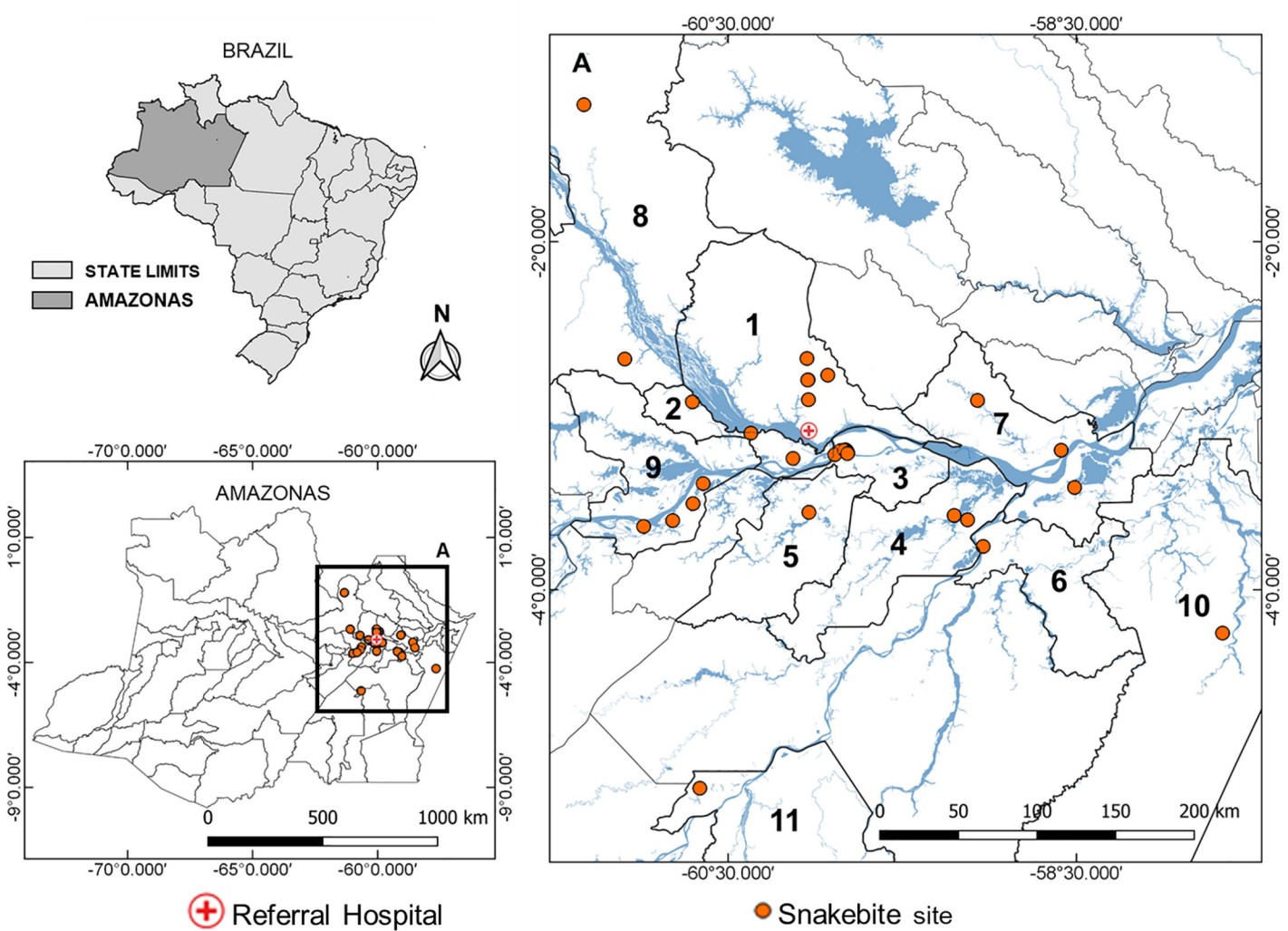

**Fig 2. Municipalities of origin of the study participants, with distance from the referral hospital.** 1) Manaus, 2) Iranduba (34 km), 3) Careiro da Várzea (37 km), 4) Autazes (232 km), 5) Careiro Castanho (132 km), 6) Nova Olinda do Norte (247 km), 7) Itacoatiara (269 km), 8) Novo Airão (192 km), 9) Manacapuru (97 km), 10) Maués (397 km), and 11) Novo Aripuanã (1378 km). The base used to create the maps is from the Brazilian Institute of Geography and Statistics, (https://portaldemapas. ibge.gov.br/portal.php#homepage).

## Themes raised from content analysis

The four main themes encountered were the exposure to snakebite during daily activities, use of traditional therapeutic practices, and personal perception of the severity of the envenomation, and the route taken and its contingencies.

**Theme 1. Exposure to snakebite during daily activities.** A very striking quote in some interviews was the feeling of safety reported by the study participants in the environment close to their homes or on the paths frequently traveled between their homes and workplaces. However, snakebite occurred remarkably close to the residence, in areas of routine transit of people, not necessarily in activities within the forest area.

> "I was going to a friend of mine's. I had even invited another friend to go with me. He even went with me, but he returned. I went on the path, on the clay path, which is the clear path..., Then, when I got close to this friend of mine I had invited to go with me that was already

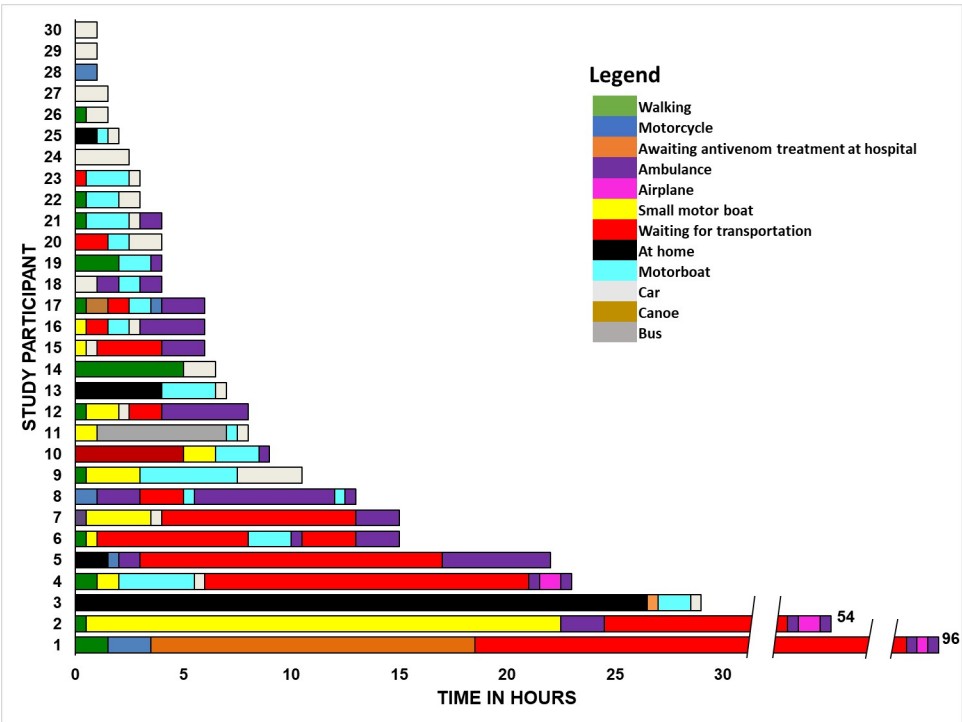

**Fig 3. Itineraries of the study participants expressed graphically from the time from the bite to hospital admission.** The different colors of the fragments of each itinerary represent the participant's stay at home after the trip, the means of transport employed, and the waiting times for transport that occurred along the route.

*ahead of me. I looked and I just saw my finger was hurt and there was a snake, so I just kicked it away."* (Participant 15: a 52-year-old brown-skinned female)

Exposure to snakebites can occur very close to the place of residence in an area routinely used by people. Although they are not engaging in an activity that places them within a forest area, the fact that they live in a rural community makes it possible for snakes to enter this environment in places that are occupied by people during common daily activities. This reality is represented in the statements below.

*'It was like this. . .it was about 9 am and we were going to where we catch saúva-tauim, but it was just me on the trail and it seemed that the animal was waiting for me there and that was when I felt the bite, like the sting from a tucandeira ant. The snake was right there on the trail."* (Participant 4: a 37-year-old brown-skinned male)

*"Because I live off fish, I had nothing to eat. So I went to repair a net that I had to fix. I was going to catch fish and bring it to my wife to cook. . .I went to get the net and that was when the snake bit me."* (Participant 11: a 46-year-old brown-skinned male)

*"I went to the tavern at night to buy candles and some incense that we put to ward off mosquitoes in the morning. On the way to the tavern, I got in the car and went to the Pau Rosa trail, but when the car stopped and I opened the door to go out, the snake was crossing the trail. I didn't see that it was close to the wheel and I stepped on it. . . ."* (Participant 18: a 38-year-old brown-skinned male)

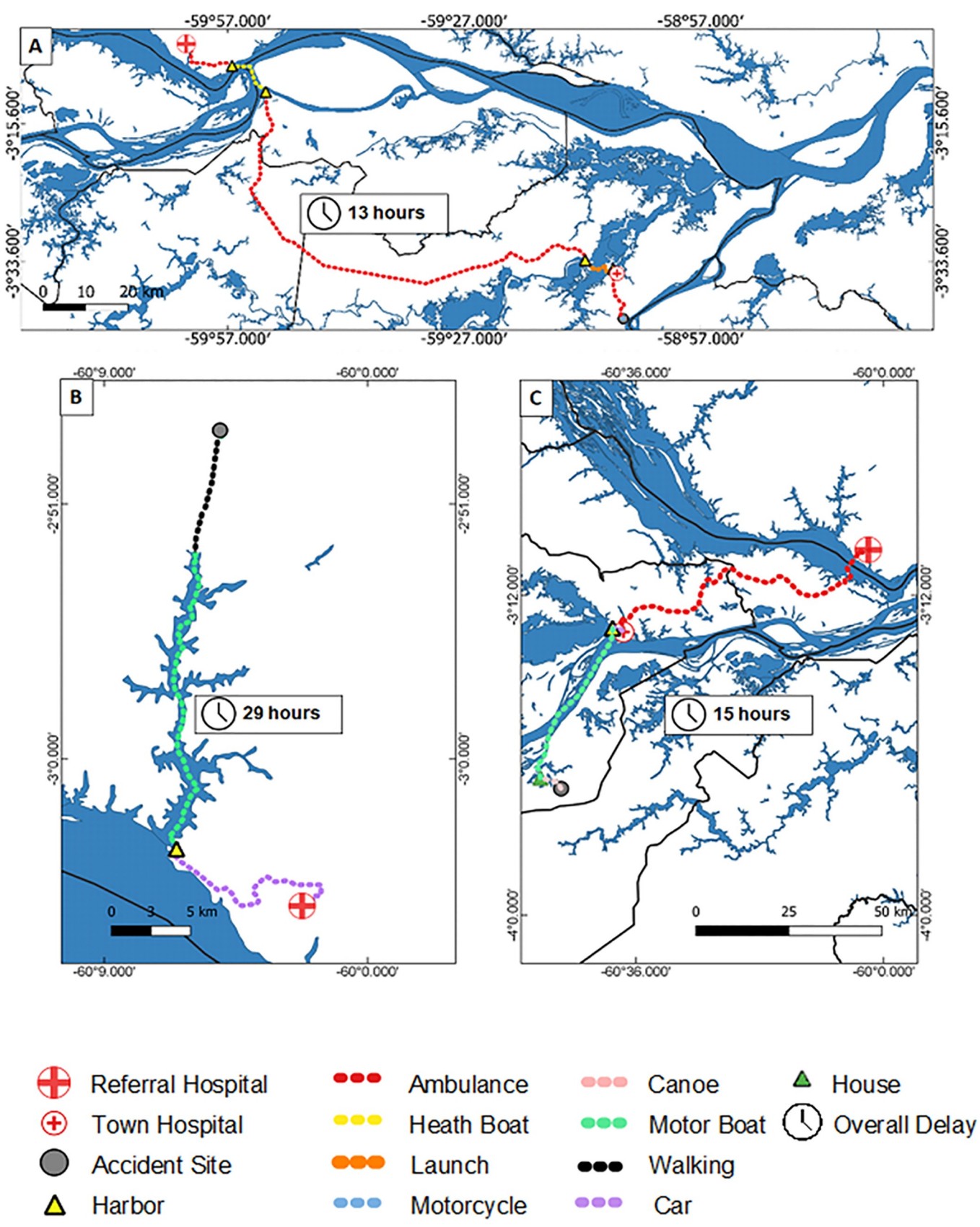

**Fig 4. Spatial visualization of the physical itineraries of three of the study participants.** A) Snakebite on a farm in the municipality of Autazes. The patient sought medical assistance shortly after the bite and had to travel in part by motorcycle to the nearest community. From there, the patient was transported by ambulance to the hospital in the city of Autazes, where he had to use four more means of transport to finally reach the referral hospital (Manaus) about 13 hours after the bite. B) Snakebite in the rural area of the municipality of Manaus. The patient sought medical assistance after 28 hours of the bite. Thinking that it was not serious, he did not inform his family regarding the envenomation. After significant worsening of local pain and edema of the left foot, which prevented him from walking, he was transported by speedboat to the urban area of Manaus and then transferred to the referral hospital approximately 29 hours after the bite. C) Snakebite in the rural area of the municipality of Manacapuru. The patient was hunting near a river when he was bitten and had to row to his house in a canoe to then get a motorboat up the river to reach the urban area of Manacapuru, where he had to wait 9 hours for an ambulance to bring him to the reference hospital in Manaus. He arrived 15 hours after the bite.

**Theme 2. Use of traditional therapeutic practices.** Procedures commonly mentioned by patients were sap extracted from plants, plant infusions, and products of animal origin (such as snake oil and gall from the *Agouti paca*). In addition, a concoction known as *Específico Pessoa* was frequently reported, which was purchased in small countryside pharmacies, though it is not registered as a medicine. Industrialized drugs were commonly kept at home, especially painkillers. Consumption of milk was claimed to reduce the symptoms of envenomation. Water consumption was contra-indicated. The use of some popular remedies or traditional and complementary medicine was also influenced by previous experiences. Some patients who had previous experience of snakebites with family members, or who had already been bitten, chose not to be submitted to any pre-hospital treatment, and showed a great concern regarding immediate medical care.

"*At the time I didn't feel anything, so I waited at the house of a woman who lives nearby while he* (a friend) *killed the snake and he tied a shirt here on my leg so the venom wouldn't travel up, just above the bite. Then he cut off the snake's tail, right? Because they say it's good, right? Because right after it bit me, I was in a lot of pain. Then he cut off the tail and put it on top of the bite, her tail there was where the pain went to at the time because they say it pulls venom from it.*" (Participant 6: a 38-year-old white female)

"*Then I didn't put anything on the wound. You don't put anything on it because it's not recommended, is it? And I heard about it from the doctors because my brother was bitten once because what he drank was still condensed milk and that was what I drank, too.*" (Participant 15: a 52-year-old brown-skinned female)

"*Then I remembered that if we tie a cloth around the limb and trap the venom, it won't spread to the rest of the body right? When I tied my leg up, a lot of people were helping. I pressed down on one side and on the other side of the knee really tightly. . .I was scared but they began to tighten it! But the doctor told me off [when he saw the tourniquet], he told me off, but in a moment of despair, you don't know what you have to do. The most important thing is my life. . .*" (Participant 18: a 38-year-old brown-skinned male)

**Theme 3. Personal perception of the severity of the moment.** The perception of the potential seriousness of the snakebite was variable, and the previous knowledge of the severity or the onset of symptoms, such as severe pain, edema and bleeding, were decisive factors for starting the journey to the hospital. In this sense, some quickly understood the snakebite to be something potentially severe that put them in a life-threatening situation, while others decided to seek help only after the signs and symptoms became very evident and uncomfortable.

Presence of signs and symptoms, especially the intense pain, was striking and was often the fact that led the individuals to seek medical treatment to resolve a situation that had reached unbearable levels of pain. Perception about potential severity was initially low and, as a result, patients only sought specialized care either after the worsening of symptoms or due to the insistence of family members and/or other people.

*"So, this is the second accident that I have to deal with here, [Though I have had] more than nine [bites]. . . I just came again because the doctor cut my leg the last time, because when I arrived at the hospital my leg was already bad. . . In the last bite, I almost lost my leg. So, I am learning."* (Participant 11: a 46-year-old brown-skinned male)

*"I really don't like seeing a doctor, but as the situation was bad and my son insisted, I came though to come at night is more dangerous (referring to the means of transport). Then I sent for my son because I was spitting pure, pure blood. I still didn't want to come but it was the only way, right? Right after the trauma, I felt a lot of pain in the hips and my head hurt, but I took my medicine for hypertension and it went away. Here, [at the hospital] I started taking the medicines that made everything stop hurting. Here, only my urine is still coming out with blood in it."* (Participant 13: a 23-year-old brown-skinned male)

*"Then, when it was about 7 o'clock at night, I couldn't take any more pain, my whole leg started to go numb and I was at my cousin's house still and I couldn't step on the floor or walk there anymore. They put me on a motorboat that they had and transferred me to Manaca-puru [. . .] I stayed at home because the people from the countryside were saying "No, this is not poisonous" and I believed them because it is something I didn't know about. Right there, when I saw that my leg was getting harder, I said "No, no, I want to go to the hospital. So I came as fast as possible."* (Participant 25: a 56-year-old brown-skinned male)

**Theme 4. The itinerary taken and its contingencies.** Availability of resources in an emergency can be decisive for a good prognosis. At the time of the snakebite, patients had extremely limited access to transport. As they were geographically isolated, some had to walk long distances until they found some means of transport. Without means of communication, they had no way to seek help, except from those who were nearby at the time of the bite.

Most of the victims depended on river transport (motor boats, boats and canoes), but sometimes the journey included land transport vehicles, such as motorcycles, cars, and even small airplanes, in order to reach the specialized health center. These means of transport were not always safe or available immediately, and patients were susceptible to weather and road conditions, as well as risk of piracy during the journey.

Despite the reference center for snakebites being well known, information on how the urgency and emergency system worked or how to get to the reference hospital was not always available and patient ended up stopping in other places that did not provide antivenom treatment. Victims needed to borrow or spend their own money to get a transfer to the correct healthcare unit. In addition, they were prone to getting lost on the trails or taking longer routes, which caused a greater delay in receiving treatment.

It was not uncommon for patients to seek medical assistance in health units without antivenoms because they were unaware of the need for specific treatment, which is not available in all clinics. Some reported the lack of availability of antivenoms even in hospitals that were supposed to receive stocks from the official supply chain. This generated a preference for treatment at the reference unit in Manaus. This, in turn, increased the length of the route to be taken and time taken.

Finally, reports of suffering during the journey deserve to be highlighted, since journeys often involved walking and uncomfortable means of transportation for a patient in great pain and limited mobility.

*"We came quickly to the city and I was on the back of the bike with another friend behind me holding me. And with every bump, my God in heaven! it hurt too much. . .the day I was supposed to come to Manaus, it rained so much and it was raining a lot, so I ended up staying*

*home. It was only on the fourth day that the weather improved, I came and it was about 8 in the morning. I came by plane and we landed here at the airport.*" (Participant 1: a 48-year-old white female)

"*We went after a more powerful boat you know, to take me to Manacapuru, but the owners there didn't want to take me at that time, afraid of the drug traffickers on the Solimões River, right*?" (Participant 6: a 38-year-old white female)

"*The ambulance driver came and then went to another community to pick up a nurse who was supposed to come. I All of this took us more than an hour in order to go to the Port of Ceasa. And I was afraid of dying, my heart squeezing and beating very hard and my pressure was getting lower and I said "Jesus I'm going to die. Don't let me die!" And this pain that doesn't stop and it seems like it's tearing my whole leg apart, and I feel like I'm all numb. It hurts, hurts, hurts, hurts, hurts! As if it were tearing me.*" (Participant 18: a 38-year-old brown-skinned male)

"*My brother-in-law was driving the car and didn't know where it was. Then we were doing the turnpike and ended up in Cidade Nova* (another neighborhood) *and so I was lost with my brother-in-law driving. Because he didn't know where it was. . . It took well over an hour.*" (Participant 20: a 56-year-old brown-skinned male)

## Late medical assistance, participants' characteristics and therapeutic itineraries

Late medical assistance was reported by seventeen (57%) participants. Gender distribution was similar in those that received late medical assistance. Indigenous participants (12%) were present only in this group. In the group with early medical assistance, there was a predominance of participants over 40 years of age; and the elderly ($\geq$60 years) were present only in this group. Schooling was slightly higher in the group with early medical assistance. Those who are married or in stable relationships slightly predominated in the group with late medical care. The participants were mostly occupied in agriculture, hunting and fishing activities, but this type of occupation was more frequent in the group with early medical assistance (Table 3).

In Fig 5, 13 variables that emerged from the interviews or from the analysis of the physical itinerary are evaluated in regards to their frequency. Firstly, it is clear that a set of variables can act simultaneously to format the final TI, pushing more or less towards a delay in treatment (Fig 5A). Most of the variables presented discrepancies between patients who arrived late to the hospital and those who received early medical assistance (Fig 5B). Feeling safe at home and in work places apparently was not a deciding factor in building the TI. Despite changing the perception of participants in relation to the demand for medical care, as observed in some statements, the frequency of a previous snakebite was small and similarly distributed between groups. In addition, perception of the seriousness of the situation experienced after a snakebite was more frequent in the group with late medical care. It shows that despite the knowledge acquired in this previous experience, and perception of risk after a snakebite, which led the participants to start their itinerary immediately, other factors determined the difficulty of access to healthcare. Similarly, lack of knowledge of the route to take to reach the hospital was more frequent in patients arriving early to the hospital, although the participants highlighted in their reports that this fact compromised them so that they did not arrive at the hospital earlier. Timely identification of warning signs/symptoms was related to early assistance. Absence of help from other people, means of transport or communication, feeling of isolation in the place of residence or work, use of traditional treatments, and exposure to snakebite in daily activities were variables that were more frequently noted in the in the group that received late

**Table 3. Late medical assistance according to characteristics of the study participants.**

| Variable | Delay for access to treatment¶ | | | |
|---|---|---|---|---|
| | Yes (n = 17) | | No (n = 13) | |
| | Number | % | Number | % |
| **Sex** | | | | |
| Male | 16 | 94 | 11 | 85 |
| **Ethnic background** | | | | |
| Mixed | 14 | 82 | 12 | 92 |
| Indigenous | 2 | 12 | 0 | 0 |
| White | 1 | 6 | 1 | 8 |
| **Age group (years)** | | | | |
| 18–29 | 7 | 41 | 3 | 23 |
| 30–39 | 6 | 35 | 2 | 15 |
| 40–59 | 4 | 24 | 6 | 46 |
| ≥60 | 0 | 0 | 2 | 16 |
| **Schooling (in years)** | | | | |
| ≤4 | 11 | 65 | 6 | 46 |
| >4 | 6 | 35 | 7 | 54 |
| **Marital status** | | | | |
| Unmarried/divorced | 7 | 41 | 7 | 54 |
| Married/stable relationship | 10 | 59 | 6 | 46 |
| **Occupation** | | | | |
| Agriculture, hunting and fishing | 13 | 77 | 7 | 54 |
| Others | 4 | 24 | 6 | 36 |

¶Medical care was delayed if the patient was treated ≥6 hours after the bite.

medical care. Living more than 200 km from the hospital, needing ≥3 means of transport during the trajectory, needing any type of river transport needed; and waiting ≥1 hour for transportation during the itinerary were also more frequent in the in the group that received late medical care (Fig 5C).

## Discussion

Narratives and statements for the understanding of TIs have been recorded mainly in urban populations with chronic or non-transmissible conditions [31–37], for HIV [38, 39] and for a few neglected diseases such as tuberculosis [40, 41], malaria [42, 43] and Buruli ulcer [44, 45]. This study describes the TIs of snakebite victims, and includes the difficulties and unforeseen circumstances that victims had to bear in order to receive antivenom treatment. Previous studies carried out in the Brazilian Amazon region have demonstrated the effect of delays in the treatment of snakebites as a risk factor for complications and even death [3, 12]. The delay in treatment is usually attributed to the great distances to be traveled by patients in order to reach a healthcare unit where antivenom is available [3, 12], or is attributed to cultural factors that would cause patients to choose traditional treatment [14]. In this study, we found that these reasons are plausible, but they only partially explain the processes by which individuals chose their paths to reach a reference hospital for snakebite treatment. In fact, our results indicate that a series of subjective factors, such as choices for self-care and ones related to the different dimensions of access to healthcare, are integrated to determine the individual trajectories adopted by each victim.

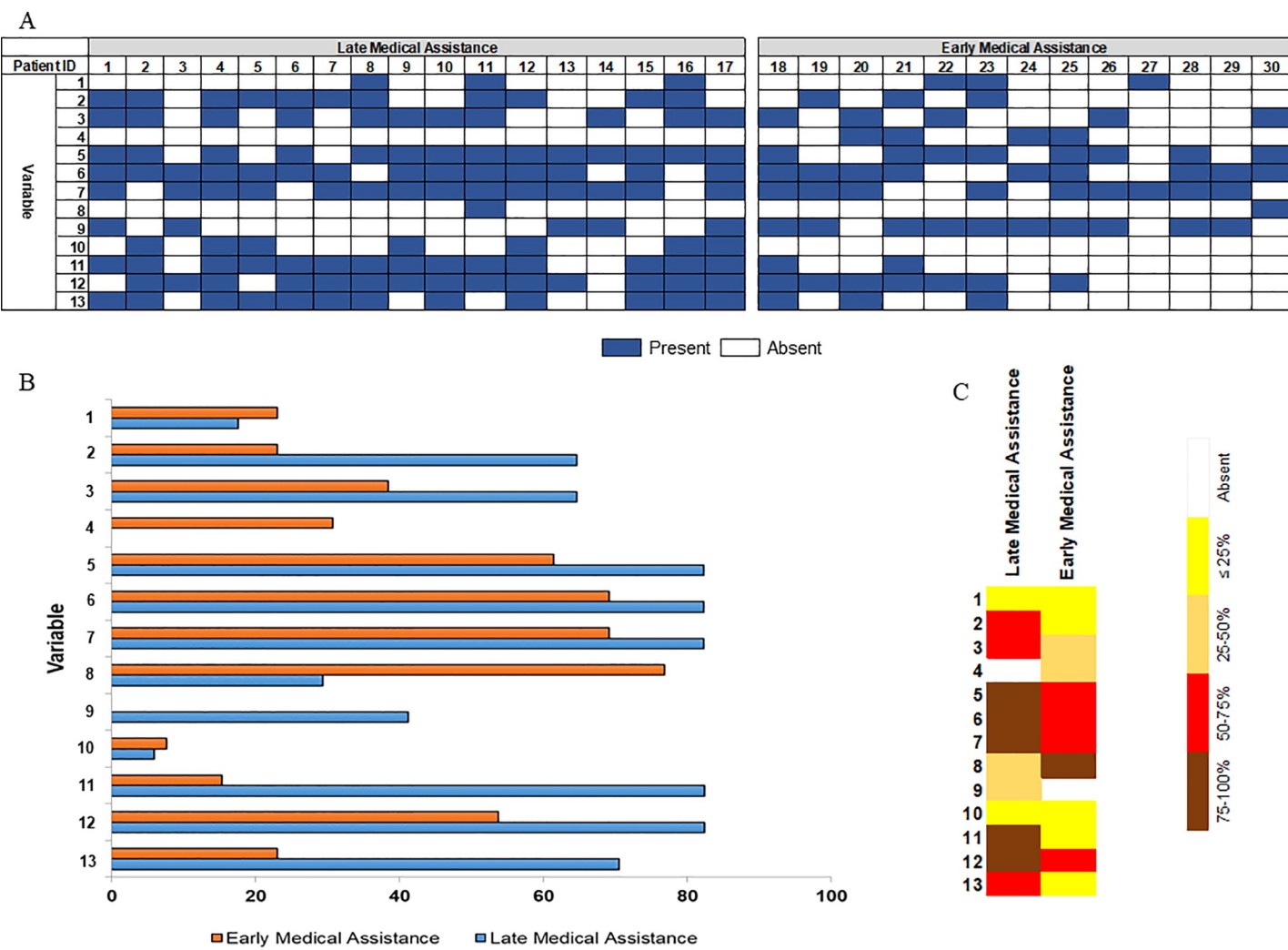

**Fig 5. Frequency of variables that emerged from the interviews or from the analysis of the physical itinerary according to late medical assistance.** 5A) Set of variables acting to format the final TI, pushing more or less towards a delay in treatment for a particular individual; 5B) Variables' frequencies between patients who arrived late to the hospital and those who had early medical assistance; 5C) Synthetic representation of frequencies between patients according to time to medical assistance. Dictionary of variables: 1. Feeling of safety at home and work places; 2. Absence of help from other people, means of transport or communication; 3. Feeling of isolation in the place of residence or work; 4. Lack of knowledge of the route to take to reach the hospital; 5. Use of traditional treatments; 6. Exposure to snakebite in daily activities; 7. Perception of the seriousness of the situation experienced; 8. Timely identification of warning signs/symptoms; 9. ≥200 km to reach hospital; 10. Previous experience with snakebites; 11. ≥3 means of transport needed; 12. River transport needed; 13. Waiting for transportation ≥1 hour over the itinerary. Variables numbered 1 to 8 emerged from the interviews' statements and variables numbered 9 to 13 derived from the physical itinerary analysis. Medical care was late if the patient was treated ≥6 hours after the bite.

## Internal structure of the health system, dimensions of access and physical itineraries

In Brazil, antivenom treatment is only available in reference healthcare units where doctors can prescribe it. Particularly, in the Amazon region, there is a shortage of medical professionals, and some municipalities have only 1 doctor per 5,000 inhabitants, which demonstrate a panorama of enormous vulnerability [46]. Thus, accessibility to antivenom treatment is deficient due to the low availability of healthcare units that offer treatment for snakebites. In the interior of the Brazilian Amazon region, each municipality usually has only one emergency care facility, which is located in its urban area, at which the patient needs to travel to in order

to receive antivenom treatment. Thus, healthcare network management in the Amazon needs to be considered not only with regards to the geographical, economic and cultural characteristics of the region, but also regarding their insufficient financing, governance and technical capacity [47, 48].

Distance as a barrier was combined with geographical barriers such as rivers and badly maintained roads, which in fact influenced some itineraries. Thus, what draws attention, especially in cases in which hospital admission was greatly delayed, is the high level of fragmentation of the itineraries, marked by several changes of mode of transport along the route, until arrival at the hospital. This makes the route time-consuming and uncomfortable for the already debilitated patient. Most study participants needed to use more than one mode of transport, alternating partly by fluvial routes and partly by terrestrial routes and even by air in some cases. As many of the fragments are unconnected with the following fragment, due to factors related to communication problems or because the next means of transport has fixed departure times, many participants had very long waiting times between one fragment and another. Regarding affordability, antivenoms in Brazil are distributed free of charge to patients. In 1986, the Brazilian Ministry of Health implemented the National Program for Snakebite Control, which was extended to other venomous animals in 1988; since then, antivenom production has been standardized and all the production from national laboratories is acquired by the Ministry of Health [10].

One factor that can affect the accessibility to antivenom is related to the costs of travel, procedures and obtaining pre- and post-hospitalization medications, as well as the loss in income due to time off work. Some patients in this study reported financial difficulties that stemmed from the need to acquire fuel for cars or boats (either their own or borrowed) expenses with taxis, medicines and food during the journey. In the Brazilian Amazon, the total cost of snakebite was estimated to be almost US$8 million in 2015, with US$ 1,539,518.62 attributed to the loss of productivity due to absence from work and US$ 268,914.18 being the cost from the patients' perspective [49]. Since victims of snakebites are generally economically vulnerable, reducing this barrier of distance by improving or providing transport systems or reducing costs for the patients through the use of cash rewards or refunds [50] may have a positive impact on snakebite victims' accessibility to health care. In southeastern Nepal, rapid transport of victims to snakebite treatment centers by motorcycle volunteers combined with simple educational messages has decreased the case fatality from snakebites rate from 10.5% to 0.5% [51]. In the Brazilian Amazon, health authorities have implemented fluvial medical units (FMUs) as an alternative in order to increase access to healthcare and healthcare coverage. However, from the users' perspective, although the presence of the FMU provides greater healthcare coverage, the financial burden generated in the pursuit for services persists. This is due to the fact that the dispersed housing pattern requires travel by users to reach the mobile unit's services, which also occurs when seeking specialized care, such as antivenom treatment; a service which is unavailable in the FMU [52]. Specific regional conditions are also mentioned in some interviews, such as poor road conditions, bad weather for air and river transport and lack of safety in night navigation due to assaults on boats.

In the state of Amazonas, of the 62 municipalities, the only one that did not have the antivenom available during this study was Careiro da Várzea, whose urban area is about 40 minutes from Manaus by river, thus all the snakebites of this locality are referred to Manaus. As such, it was expected that the cases included in the study would come only from Manaus and Careiro da Várzea, as well as some serious cases from any other municipality. However, it was observed that patients seek care directly in Manaus, even those who live in places located between 200 to more than 1,000 kilometers from Manaus by land or river routes, which contribute late medical assistance. As a result, although many patients start their individual

itineraries to the hospital immediately after the bite, which reveals an understanding of the need for professional treatment by this population, they often make mistakes regarding their itineraries, preferring care at the reference hospital, despite the greater distance to be traveled. We strongly believe this behavior is related to the acceptability and accommodation dimensions. Some patients report that the referral service in the capital is in fact sought as a priority. Perhaps these results are related to previous frustrations due to the low resolution of health problems in their municipalities of residence, as observed for other health problems [53–55]. Stories of previous snakebites with an unfavorable outcome in these places, better accommodation in the reference unit, or even symbolic issues that raise the reputation of the capital's health services in relation to small inland municipalities, may also be involved.

In view of the above, in order for a decentralization program of antivenom treatment to health facilities of lower complexity to be effective, the program must consider aspects of physical infrastructure and the different components involved in acceptability. In Ecuador and Tanzania, successful management of snakebite envenomations was achieved in a severely resource-constrained area by improving access to snakebite treatment in nurse-led clinics [56, 57]. Initiatives in this sense need to be urgently evaluated in the Brazilian Amazon.

## Self-care, subjective domains and itinerary choices

The housing and areas where subsistence working is carried out in the rural areas were considered as safe by the participants. As these places generally do not have undergrowth and are cleaned periodically, with no accumulation of garbage, leaves and branches, the participants believe that the snakes will not frequent them. In this environment, participants feel safe enough not to take any protective measures, walking barefoot most of the time. However, the snake *Bothrops atrox*, responsible for about 90% of snakebites in the Amazon region, is the most abundant species in forested areas of the region [58, 59], and also inhabits areas close to disturbed habitats around human settlements, including pastures and crops, and urban areas [15, 59, 60]. Harvesting of palm fruits and other forest products is a conducive activity that is linked to a higher risk of snakebites by *B. atrox*; palm trees may attract rodents, which feed on the fruits that fall to the ground or are on the tops of trees, serving as a food source for both adult and juvenile snakes, respectively [61]. Previous studies have shown that most of the *B. atrox* specimens that caused bites were classified as juveniles [62]. Since juveniles of this species are arboreal, snakebites also occur in upper limbs, as observed in this study. Thus, as lower and upper limbs are the most affected areas of body, the use of boots, leggings and gloves are recommended as the main primary prevention measures. Educational programs about safe habits for the most vulnerable groups are also essential.

In the Brazilian Amazon, first aid measures that would not be medically effective for the victim's recovery are commonly utilized [14]. In the self-care practices reported by the study participants, many procedures that can contribute to the worsening of the local clinical picture of the envenomation are observed. As in African settings, participants generally recognized the need for antivenom treatment, but had inadequate knowledge about appropriate first aid methods [63]. Practices such as the use of tourniquets, incisions in the affected area, and use of substances of several origins are believed by the patients as a way to remove venom or to prevent its spread through the body. However, such procedures are contra-indicated due to risk of necrosis and secondary bacterial infection. Traditional oral medicines, such as plant or animal-derived preparations, and the remedy known as *Específico Pessoa* were commonly reported. Painkillers, such as metamizol, were also commonly used. In the reports of the patients, it is possible to observe that the search for pain relief in order to endure the journey they will have to face travelling to the hospital, is an important reason for the use of traditional

therapeutic strategies. These treatments are sometimes considered an alternative to going to the hospital, or at other times a form of relieving symptoms during the journey. The knowledge of these medicines comes from the family itself, handed down from the parents, or in some cases from neighbors and shamans. Even though some patients do not see much plausibility in the effectiveness of these treatments themselves, they understand that this is a necessary resource, which could eventually be beneficial during their journey to the hospital. In fact, traditional healing practices are common in many cultures, and some communities report that even with access to medical care, traditional healing practices are still used [64]. Use of these ineffective or even deleterious self-care practices are recorded across the world as being the cause of late medical assistance and poor prognosis in snakebites [14, 63, 65]. Interestingly, however, some participants who had a previous experience of snakebite in the family, or had previously been bitten, decided not to use traditional measures, as they had been discouraged to do so by health professionals on previous occasions. Awareness of the need for antivenom treatment provides a good starting point for mass educational campaigns that take advantage of the knowledge of these "local agents", who should share previous experiences.

The way snakebite patients recognize their symptoms as being potentially severe enough to seek medical attention was clearly key to understanding how they make a decision to start the itinerary. The perception of the potential severity of the snakebite varies among the participants and, as in other deprived groups [66], some of them sought help only after warning signs, such as the onset of unbearable pain, disfiguring edema, bleeding and decreasing functional mobility. In socio-economically disadvantaged populations, these warning signs may be downgraded in importance due to a lack of a positive conceptualization of health, and normalization of the clinical condition [67]. On some occasions, healthcare service utilization by snakebite victims was a reflection of the participant's resistance to seeking medical assistance, and this was only overcome by pressure from family members. When participants understand that snakebite is a life-threatening situation, they often seek help more quickly, since they are afraid of leaving their relatives without financial support in the event of their demise. In our study, schooling was slightly higher in the group with early medical assistance. We did not specifically investigate the effect of this variable on the time to care, but possibly more educated participants tend to be more informed about the procedures to be adopted after the bite, have more financial resources for transportation to the hospital, and may have greater interaction and ability to articulate in the community, facilitating access to the health system. Likewise, the interviews indicate that older people sought medical help more readily because of their own previous experience with snakebites or with family members who have suffered the same problem. In addition, these individuals often have a higher income than other members of the community, from pensions.

## Limitations

One of the limitations of the present study is inherent in any investigation based on interviews, in which the participants' affirmations may be impregnated with interests, hesitations, incongruities, strategies and conflicts, and, as a result, may affect the final analysis in a way that the researchers cannot control. The results obtained here reflect the reality of the recruited individuals, with their own identity and culture, and which are built in their community, and do not have the intention of being extrapolated to other human groups in the Amazon. It should be noted that in this study only the reports of individuals who felt compelled to make the journeys from the accident site to the hospital where the antivenom was available were analyzed. A large proportion of individuals may not be able to obtain the treatment offered, and their reports may contain elements that are different from those obtained here. For example, in

remote communities in the Juruá and Solimões River basins, the proportion of riverside residents who do not seek hospital care is around 75 and 39%, respectively [68]. Future studies with these specific populations may reveal strategies for coping with a problem that is even more impregnated with subjectivity in self-care and a greater participation of the folk sector as the only healthcare choice.

## Conclusion

Therapeutic itineraries and receipt of healthcare is the outcome of many different complex processes. Access to proper healthcare requires considerable effort on the part of snakebite victims, and the quantity, difficulty, and complexity of this effort serve as obstacles to the receipt of proper care. Major obstacles identified from the in-depth analysis were 1) Low availability of healthcare units that offer antivenom treatment, 2) Poor accessibility to healthcare, related to long distances, geographical barriers, such as rivers, transport shortage and associated costs and badly maintained roads, which results in greater fragmentation of itineraries, 3) Possible low acceptability of the healthcare currently offered in the countryside at the expense of the service offered at the reference unit in the capital, 4) In households and workplace areas, individuals feel safe and do not perceive the need to use personal protective equipment, 5) Participants generally recognized the need for antivenom treatment, but had inadequate knowledge about appropriate first aid methods, and commonly use ineffective or even deleterious self-care practices, 6) Timeliness of healthcare utilization by snakebite victims was the result of the balance of participant's ability to recognize their symptoms as being potentially severe, and resistance to seeking medical assistance. Therefore, public health interventions should focus on health education and on improving victims' rapid access to antivenom by promoting immediate and fast transport to adequate treatment centers, particularly for bites occurring at night. More strategically-placed antivenom distribution points among existing community healthcare centers in the Brazilian Amazon may be a solution that maximizes coverage and minimizes the time to care.

## Supporting information

**S1 Text. Transcription of the interviews.**
(DOCX)

## Author Contributions

**Conceptualization:** Vinícius Azevedo Machado, Fan Hui Wen, Wuelton Marcelo Monteiro, Jacqueline Almeida Gonçalves Sachett.

**Data curation:** Joseir Saturnino Cristino, Guilherme Maciel Salazar, Altair Seabra Farias.

**Formal analysis:** Joseir Saturnino Cristino, Guilherme Maciel Salazar, Vinícius Azevedo Machado, Eduardo Honorato, Altair Seabra Farias, Alexandre Vilhena Silva Neto.

**Funding acquisition:** Marcus Lacerda, Wuelton Marcelo Monteiro, Jacqueline Almeida Gonçalves Sachett.

**Investigation:** Joseir Saturnino Cristino, Guilherme Maciel Salazar, Vinícius Azevedo Machado.

**Methodology:** Joseir Saturnino Cristino, Vinícius Azevedo Machado, Eduardo Honorato, Altair Seabra Farias, João Ricardo Nickenig Vissoci, Marcus Lacerda, Fan Hui Wen, Wuelton Marcelo Monteiro, Jacqueline Almeida Gonçalves Sachett.

**Project administration:** Vinícius Azevedo Machado, Wuelton Marcelo Monteiro, Jacqueline Almeida Gonçalves Sachett.

**Resources:** Vinícius Azevedo Machado, Eduardo Honorato, Marcus Lacerda, Fan Hui Wen, Wuelton Marcelo Monteiro, Jacqueline Almeida Gonçalves Sachett.

**Software:** Joseir Saturnino Cristino, Alexandre Vilhena Silva Neto.

**Supervision:** Wuelton Marcelo Monteiro, Jacqueline Almeida Gonçalves Sachett.

**Validation:** Guilherme Maciel Salazar, Vinícius Azevedo Machado, Eduardo Honorato, Altair Seabra Farias.

**Visualization:** Joseir Saturnino Cristino, Guilherme Maciel Salazar, Alexandre Vilhena Silva Neto, Wuelton Marcelo Monteiro.

**Writing – original draft:** Joseir Saturnino Cristino, Vinícius Azevedo Machado, Wuelton Marcelo Monteiro, Jacqueline Almeida Gonçalves Sachett.

**Writing – review & editing:** Eduardo Honorato, Altair Seabra Farias, João Ricardo Nickenig Vissoci, Alexandre Vilhena Silva Neto, Marcus Lacerda, Fan Hui Wen.

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
