## [Decision Letter · Decision Letter 0]

29 Jan 2021

Dear Dr Monteiro,

Thank you very much for submitting your manuscript "A painful journey to antivenom: the therapeutic itinerary of snakebite patients in the Brazilian Amazon (The QUALISnake Study)" for consideration at PLOS Neglected Tropical Diseases. As with all papers reviewed by the journal, your manuscript was reviewed by members of the editorial board and by several independent reviewers. The reviewers appreciated the attention to an important topic. Based on the reviews, we are likely to accept this manuscript for publication, providing that you modify the manuscript according to the review recommendations. 

All reviewers picked up on editorial issues that will require attention - including spelling, sections exceeding journal guidelines on length and relating to the resolution of images. Please consult the guide for authors regarding the latter two issues. 

Sincerely,

Stuart Robert Ainsworth

Associate Editor

José María Gutiérrez

Deputy Editor

Reviewer's Responses to Questions

**Key Review Criteria Required for Acceptance?**

**Methods**

-Are the objectives of the study clearly articulated with a clear testable hypothesis stated?

-Is the study design appropriate to address the stated objectives?

-Is the population clearly described and appropriate for the hypothesis being tested?

-Is the sample size sufficient to ensure adequate power to address the hypothesis being tested?

-Were correct statistical analysis used to support conclusions?

-Are there concerns about ethical or regulatory requirements being met?

Reviewer #1: This is a qualitative study, based on the in-depth interviews for describing the therapeutic itineraries. The sample size has been defined based on the data saturation.

Reviewer #2: (No Response)

Reviewer #3: The study population was representative of opinion from several different municipalities. The cross-sectional, qualitative methodology was appropriate to the objectives, as was the data analysis. The rationale for this study and its methodology was clearly described.

The sample size seemed low to me - especially given the number of variables assessed (particularly in Figure 5).

I would like to have understood why children under 18 years were excluded - this cohort is often the most frequently bitten by snakes and typically suffer more, and more rapidly, than adults. A brief rationale for this decision would be valuable. 

I would like to have understood better the influence of schooling, access to snakebite care information (media, lay knowledge) had on perceptions of conventional hospital care and the decisions made to access it.

How were bites ascribed so robustly to Bothrops atrox (100%)? Was some kind of diagnostics available? A brief explanation would be valuable.

The authors and their team deserve congratulations for this physically difficult study and one requiring considerable sensitivity to acquire the detail obtained. It would be informative for other studies being planned to learn whether they experienced issues over participant expectations following the interviews - financial assistance, travel assistance, treatment, etc? if so, how were these managed?

**Results**

-Does the analysis presented match the analysis plan?

-Are the results clearly and completely presented?

-Are the figures (Tables, Images) of sufficient quality for clarity?

Reviewer #1: The four themes identified are appropriate and are and described adequately. They are adequately supported with the quotes. They are well-supported with the figures and tables.

Reviewer #2: (No Response)

Reviewer #3: The analysis was performed as planned and the results were clearly and informatively presented.

I think some of the figures (at least those in the manuscript available to me) lacked the resolution required for publication - for editor's attention.

I'd have been interested in an explanation why the elderly cohort accessed healthcare sooner than the other groups.

**Conclusions**

-Are the conclusions supported by the data presented?

-Are the limitations of analysis clearly described?

-Do the authors discuss how these data can be helpful to advance our understanding of the topic under study?

-Is public health relevance addressed?

Reviewer #1: The six conclusions arrived are well-justified and the limitations are adequately highlighted.

Reviewer #2: (No Response)

Reviewer #3: This is a very valuable manuscript and addresses a paucity of research on the (i) perceptions and prior knowledge of rural snakebite victims, (ii) how this influences their healthcare-access decision making and (iii) the substantial barriers they face in accessing healthcare when they chose to do so. The very fragmented travel routes to healthcare was a very important delivery of this study. A cost-implication analysis would have been a valuable addition (but you can't do everything).

As in the few other studies of this genre, this research identifies that victims often do understand bite severity and that this influences decision as to whether, or not, to seek hospital care. It was very instructive to learn that (i) that snakebite victim ITs often bypassed more local hospitals in favour of referral hospitals and (ii) that the vast majority of local health facilities had adequate stocks of antivenom.

**Editorial and Data Presentation Modifications?**

Reviewer #1: Terminology: ‘snake poisoning’ is a wrong term, it must be replaced with “snake envenoming”

Line 354: remove additional comma

Throughout the manuscript (in the text and the tables), the percentages of patients have been expressed with the precision of one decimal point. This study includes only 30 (<100) participants. Therefore, it is not accurate to express percentages using decimal points – please remove the decimal values.

After each quote, it would be appropriate to include the age, sex and ethnic background of the participant. 

e.g.: ““We went after a more powerful boat you know, to take me to Manacapuru, but the owners there didn't want to take me at that time, afraid of the drug traffickers on the Solimões River, right?” (Participant 6: a 36 year old indigenous male)

Reviewer #2: (No Response)

Reviewer #3: While this is a very well written, referenced and balanced manuscript, some minor issues need attentions:

The abstract and Author summary far exceed journal guidelines.

Numerous spelling mistakes need editorial attention.

**Summary and General Comments**

Reviewer #1: The study titled “painful journey to antivenom: the therapeutic itinerary of snakebite patients in the Brazilian Amazon (The QUALISnake Study)” by Cristino et. al. describes the therapeutic itineraries of snakebite victims in Brazillian Amazon. Understanding the socio-epidemiology of snakebite, specially on the treatment seeking aspects in most vulnerable communities is of highest importance in confronting this neglected tropical disease. As such, the most important approach would be to use qualitative approach, which is very rare among snakebite literature. In this excellent manuscript, the authors have taken a qualitative approach, based on the in-depth interviews, in describing the therapeutic itineraries. I much enjoyed reading this manuscript.

Reviewer #2: 1. Snakebite is a neglected fatal disease among the poorest and among isolated communities. Anti-venom is the only hope for saving life at snake envenoming. According to the authors, the objective of this study is to identify the barriers in access to treatment and recommend improvements need in policies/strategies to handle the situation, by analyzing the therapeutic itineraries of snakebite victims. This area of research is important and relevant to the scope of the Plos-NTD. Thanks for researchers for selecting this topic to bring the attention of the world to understand the treatment access to snakebites in Amazon region. 

2. This is a qualitative research under four thematic topics (1.Activity during exposure to bite, 2. Home remedies, 3.Knowledge of the snakebite injury and treatments, 4. Itinerary bite to treatment). Study sample consists of 30 study participants. From the statistical point of view, study design make sense to me. The objective of the study line 237 is clear to me but I feel researchers should give further attention in text to form the research question more clearer.

3. I am not fully agree with the use of term “therapeutic itinerary (TI)” in this article because it is more appropriate for an illness that may not much time sensitive. Snakebite envenoming is a life threating injury need immediate medical attention within few hours of the incidents unless or otherwise it is a dry bite or non-venomous animal bite. The all therapeutics described in this manuscript are temporary home remedies. I suggest, sometimes “Itinerary from bite to treatment” would be more meaningful in this context than TI.

4. Line 152-153; Sentence needs to improve and “preferably in the first hours after the bite” should be specific. May studies suggest within first 6 hours of the bite (with a reference).

5. Line 172-173; I guess “delay for treatment” would give a better meaning to the phrase instead ‘late medical care’.

6. Introduction of this article is very long to me to grasp the story line. Authors can consider, details given in lines 188-227 to further summarized or move to the discussion or present in an appendix.

7. Line 284; ‘late medical attention’ or ‘delay for access to treatment’ is sometimes more meaningful than ‘late medical assistance’? 

8. Line 293-294; need more clarity of ‘what kinds of inconsistencies ’ in this statement. 

9. Is it possible to elaborate the categories in table 3 to add ‘Distance traveled (KM) reach to the hospital ’, ‘mode of transport’, ‘alternative treatments before deciding hospital option ’ etc? by summarizing the numbers that have been presented in Fig 2, 3 and 5. 

10. I understand authors are not native English speakers. The English language use in Lines 350- 550 is confusing not well written. 

11. Line 577; 1 doctor per 5,000 would be more appropriate. You can’t have a 0.2 doctors.

12. Does Brazil has a public funded emergency ambulance system like 911 in US, 118 in India? Snakebite apps like in India? Those save many lives ( https://doi.org/10.1016/S2352-3018(18)30007-9)

13. I had some limitations in this review. I am sorry my inability to check some of the non-English references, as I am not familiar in languages other than in English.

Reviewer #3: I think this study is an outstanding contribution to our knowledge and one that the should help inform remedial interventions implemented by national, regional, NGO and community group stakeholders.

PLOS authors have the option to publish the peer review history of their article (what does this mean?). If published, this will include your full peer review and any attached files.

Reviewer #1: No

Reviewer #2: No

Reviewer #3: No
---

## [Editor Report · Decision Letter 1]

15 Feb 2021

Dear Dr Monteiro,

We are pleased to inform you that your manuscript 'A painful journey to antivenom: the therapeutic itinerary of snakebite patients in the Brazilian Amazon (The QUALISnake Study)' has been provisionally accepted for publication in PLOS Neglected Tropical Diseases.

Best regards,

Stuart Robert Ainsworth

Associate Editor

José María Gutiérrez

Deputy Editor

---

## [Editor Report · Acceptance letter]

26 Feb 2021

Dear Dr. Monteiro,

We are delighted to inform you that your manuscript, "A painful journey to antivenom: the therapeutic itinerary of snakebite patients in the Brazilian Amazon (The QUALISnake Study)," has been formally accepted for publication in PLOS Neglected Tropical Diseases.

Best regards,

Shaden Kamhawi

co-Editor-in-Chief

Paul Brindley

co-Editor-in-Chief
